# Reproducibility Crossroads: Impact of Statistical Choices on Proteomics Functional Enrichment

**DOI:** 10.3390/ijms26189232

**Published:** 2025-09-21

**Authors:** Karolina A. Biełło, José V. Die, Francisco Amil, Carlos Fuentes-Almagro, Javier Pérez-Rodríguez, Alfonso Olaya-Abril

**Affiliations:** 1Department of Biochemistry and Molecular Biology, University of Córdoba, Campus de Rabanales, 14071 Córdoba, Spain; karolinabiello@gmail.com; 2Department of Genetics-ETSIAM, University of Córdoba, Campus de Rabanales, 14071 Córdoba, Spain; z42diraj@uco.es; 3Central Research Support Service (SCAI), Bioinformatics Unit, Rabanales de Campus, 14071 Córdoba, Spain; b72amruf@uco.es; 4Central Research Support Service (SCAI), Proteomics Unit, Rabanales de Campus, 14071 Córdoba, Spain; b72fualc@uco.es; 5Department of Computer Science and Artificial Intelligence, Campus de Rabanales, 14071 Córdoba, Spain; javier.perez@uco.es

**Keywords:** proteomics, meta-analysis, functional enrichment, statistical methods, reproducibility

## Abstract

Quantitative proteomics relies on robust statistical methods for differential expression, critically impacting downstream functional enrichment. This meta-analysis systematically investigated how statistical hypothesis testing approaches and criteria for defining biological relevance influence functional enrichment concordance. We reanalyzed five independent label-free quantitative proteomics datasets using diverse frequentist (*t*-test, Limma, DEqMS, MSstats) and Bayesian (rstanarm) approaches. Concordance of Gene Ontology (GO) and KEGG pathways was assessed using Jaccard indices and correlation metrics, grouping comparisons by statistical test and biological relevance consistency. The results demonstrated highly significant differences in similarity distributions among the comparison groups. Comparisons varying only hypothesis testing methods (with constant relevance criteria, FC or Bayesian) showed the highest consistency. Conversely, comparisons with differing biological relevance criteria (or varied methodological choices) yielded significantly lower consistency, highlighting this definition’s critical impact on GO term overlaps. KEGG pathways displayed more uniform, method-insensitive concordance. Sensitivity analysis confirmed the findings’ robustness, underscoring that methodological choices profoundly influence functional enrichment outcomes. This work emphasizes the critical need for transparency and careful consideration of analytical decisions in proteomics research to ensure reproducible and biologically sound interpretations.

## 1. Introduction

The “omics” era has revolutionized our understanding of biological systems, with quantitative proteomics emerging as an indispensable tool for deciphering cellular complexity, discovering biomarkers, and elucidating pathological mechanisms [1]. Mass spectrometry (MS)-based proteomics has become the gold standard for large-scale protein identification and quantification. Quantitative MS workflows are typically classified according to two major dimensions: the quantification strategy (label-based vs. label-free) and the acquisition method (data-dependent vs. data-independent acquisition).

Label-based quantification, including metabolic labeling (e.g., Stable Isotope Labeling by Amino Acids in Cell Culture, SILAC), chemical labeling (e.g., Isobaric Tags for Relative and Absolute Quantitation, iTRAQ), and isobaric tagging (e.g., Tandem Mass Tags, TMT), reduces technical variability by allowing multiplexed measurements in a single run. These methods improve relative quantification but can be costly, limit multiplex capacity, and suffer ratio-compression due to co-isolated precursors [2]. By contrast, label-free approaches (spectral counting or MS1 intensities) are more scalable and cost-efficient [3] but are typically more sensitive to run-to-run variability; appropriate normalization and alignment are therefore essential to obtain robust results.

Traditionally, MS data acquisition has relied on Data-Dependent Acquisition (DDA), where the most intense precursor ions are selected for fragmentation. While powerful, DDA suffers from run-to-run variability and stochastic sampling, which can lead to missing data. In response, Data-Independent Acquisition (DIA) has emerged as a compelling alternative. In DIA, all precursor ions within defined m/z windows are fragmented systematically, improving reproducibility and proteome coverage across samples. Combined with spectral libraries or recent library-free algorithms, DIA workflows enable deep, consistent quantification in both label-based and label-free contexts.

In parallel to these quantification strategies, protein identification remains a cornerstone of proteomics. Most workflows rely on database-driven peptide matching, but de novo sequencing—as implemented in tools like PEAKS—can enhance identification in cases where reference databases are incomplete or sample-specific variants are expected. Nonetheless, all these approaches face challenges: the precision and accuracy of protein quantification are highly sensitive to various experimental and technical variables, which can introduce noise and affect the reliability of biological conclusions. Identifying and mitigating these sources of variability is therefore essential to ensure biologically meaningful and reproducible results [4].

Variability originates from multiple stages. The pre-analytical stage is critical; sample type, quality and protein lysis/extraction methods significantly influence recovery and representativeness. Different extraction protocols, for instance, often yield low overlap in identified proteins due to selective solubilization, leading to biased profiles [5]. At the protein level, intrinsic characteristics like size, hydrophobicity, and post-translational modifications (PTMs) impact proteolytic digestion, solubility, and ionization efficiency. Peptide amino acid sequence also directly influences fragmentation and thus identification/quantification efficiency [6]. The instrumental stage is influenced by ionization efficiency and peptide ion mobility, among others. Optimization of mass spectrometer acquisition parameters (e.g., injection time, resolution, collision energy), co-elution, interference, and the quality/stability of the liquid chromatogram are also crucial [6]. Poor chromatographic reproducibility or high background noise severely compromise accurate peptide quantification.

After mass spectra acquisition, bioinformatic decisions profoundly alter results [7]. The choice of protein sequence database and its comprehensiveness is fundamental since an incomplete database leads to missed protein identifications. Customized databases are increasingly vital in proteogenomics, improving identification rates for organisms with incomplete genomes or specific genetic variants [8]. Databases should be complete, up-to-date, and include relevant isoforms and known variations. Search parameters in engines are equally decisive, based on enzymatic digestion (e.g., trypsin), PTMs, and expected mass errors [9]. Strict mass tolerances can omit valid identifications, while lax ones increase false positives. These parameters directly affect peptide detection, identification, and subsequent protein quantification. Mass spectrometry search engines (e.g., Mascot, Sequest, Proteome Discoverer, MaxQuant, Comet/X! Tandem) use distinct algorithms. While often yielding similar results, they differ in sensitivity and specificity, particularly for low-abundance or complex peptides [10]. Engine choice and parameter optimization influence PSM (Peptide-Spectrum Match) identification quantity and quality. Later, peptide-to-protein inference is a non-trivial next step, requiring critical decisions to avoid protein over-identification. It involves grouping PSMs corresponding to unique peptides to infer protein presence; therefore, proteotypic peptides are crucial for unambiguous protein identification [11]. Parsimony principles and handling shared peptides among multiple proteins are key, often forming inferred protein groups. Setting FDR (false discovery rate) thresholds, typically 1% at the peptide and/or protein level, is critical for controlling false positives and ensuring identification confidence [9].

Quantitative intensity data derived from PSMs require normalization to correct for technical variability [12]. Various methods exist for intra-replicate (e.g., total chromatogram intensity) and inter-replicate (e.g., median of total peptide intensities, quantile normalization) correction. More sophisticated methods like LOESS or VSN normalize for variance-intensity dependencies [12]. Label-free specific methods like iBAQ and LFQ (in MaxQuant) perform internal normalizations across replicates [13,14]. The choice of normalization significantly impacts downstream results and the detection of biological changes [15].

Once reached, determining significantly changing proteins requires careful experimental design and appropriate statistical approaches for differential expression analysis. Designs range from simple two-group comparisons to complex multifactorial or time-series experiments (Table 1). While conventional parametric methods like Student’s *t*-test and ANOVA are widely used, their assumptions (normal distribution, homoscedasticity, independence) are often violated in proteomics due to variability, missing values, and heterogeneous measurement error [16]. These classical methods limit statistical power and increase false positive/negative rates, especially in low-replication designs.

More robust alternatives address these limitations. Limma performs moderated *t*-tests using empirical Bayes shrinkage, improving variance stability with low replicates [17]. DEqMS models protein-level variance dependence on identified peptides for precise estimates [18]. Bayesian methods (e.g., BDiffProt, BNIH) encode uncertainty and incorporate prior information, improving false discovery rate control and effect size estimation under non-normal conditions [19]. For longitudinal and multifactorial designs, linear mixed-effects models (MSstats) control for intra-subject correlation, repeated measurements, and batch effects [13]. These models, combined with variance moderation (Limma, DEqMS), outperform classical methods in high-dimensional data with missing values or low replication [21]. Beyond statistical significance, biological relevance must be evaluated. While *p*-values indicate probability, effect size or fold change (FC) quantifies the magnitude of difference, directly indicating biological relevance. Bayesian approaches offer direct inference about effect magnitude and the probability of biologically relevant differential expression, often using a Null Interval of Relevance for more intuitive interpretation [22].

The quantitative proteomics workflow lead to the performance of enrichment analyses to transform data into interpretable biological knowledge. These analyses identify disproportionately represented biological functions, processes, or pathways within lists of quantitatively changed proteins by integrating proteomic information with databases like GO (Gen Ontology), KEGG (Kyoto Encyclopedia of Genes and Genomes), and Reactome. This provides a high-level view of underlying molecular mechanisms. The robustness of enrichment results could be intrinsically linked to methodological decisions made throughout the proteomic workflow. Variability from sample preparation, data acquisition, normalization, missing value imputation, and differential expression analysis propagates, could affect input protein lists for enrichment and thus pathway interpretation. Understanding factors biasing or affecting enrichment consistency is essential for reliable biological conclusions. This study precisely addresses this fundamental need. Through a meta-analysis using the Jaccard similarity coefficient on real proteomic datasets, we aim to quantify and understand how different methodological decisions influence the robustness and reproducibility of pathway enrichment results (Figure 1). This approach will empirically illuminate how selections within the quantitative proteomics bioinformatics pipeline directly affect biological interpretation, providing a basis for optimizing workflows and enhancing confidence in biological inferences.

## 2. Results

Quantitative proteomics experiments aim to identify and quantify changes in protein abundance across different biological conditions. A critical downstream step involves pathway and functional enrichment analysis, which translates lists of differentially expressed proteins into biologically meaningful insights. The statistical methods employed for differential expression analysis can vary significantly, broadly categorized into frequentist approaches (e.g., *t*-tests, ANOVA, linear models) and Bayesian methods (e.g., typically incorporating prior information or empirical Bayes). The choice of method could profoundly impact the resulting list of significant proteins, consequently affecting the outcome of subsequent enrichment analyses.

In this work, five previously published quantitative proteomics independent studies (“Works”) were reanalyzed to elucidate the impact of the non-biological component of sample-to-sample comparison experiments using label-free quantitative proteomics. All were analyzed using the same parameters in MaxQuant (Appendix A), thus limiting the differences to the biological parameters of the experiment itself and those derived from the statistical decisions under study. Principal Component Analysis (PCA) was performed to explore the overall variance among samples and visually assess sample clustering (Appendix A).

### 2.1. Characterization of Dataset Variability and Its Influence on Proteomic Outcomes

Prior to examining the differential expression results, we analyzed the variability within each experimental condition (Table 2 and Appendix A). The median coefficient of variation (CV) offers insights into the reproducibility of protein quantification within each condition. CV revealed disparities in reproducibility across conditions. For instance, in Work 4, Cond2 showed a considerably higher CV (31.95%) than Cond1 (18.62%), while in Work 5, Cond1 had a higher CV (36.42%) compared to Cond2 (31.99%). In contrast, Works 1 and 3 displayed relatively low and similar median CVs between conditions, indicating more consistent quantification.

Levene’s test assesses the equality of variances between the two conditions. This test revealed significant variance differences in Works 3 (*p* = 0.0221) and 4 (*p* < 0.0001), but not in Works 1, 2, and 5 (0.3694, 0.9991, and 0.1310, respectively). The PCA plots (Appendix A) further illustrate the spread and variability among datasets.

### 2.2. Differential Expression Analysis: The Role of Hypothesis Testing Methods in Protein Identification

Differential expression was assessed with six methods: Student’s *t*-test [23], Welch’s *t*-test [24], Limma [17], DEqMS [18], MSstats [13], and a Bayesian approach implemented in rstanarm [25]. Although brms was initially considered for the Bayesian analysis, it proved computationally infeasible for routine use, leading to the adoption of rstanarm, which achieves greater efficiency by frequently leveraging pre-compiled Stan code [26]. Both packages are widely applied in quantitative proteomics for differential expression due to their ability to provide posterior distributions, quantify uncertainty, and incorporate prior knowledge [27]. The number of differentially expressed proteins (DEPs) varied substantially across methods and datasets (Figure 2A). For example, MSstats consistently identified the highest DEP counts in Workflows 1 and 4, whereas the Bayesian method was the most sensitive in Workflows 2 (61 DEPs), 3 (832 DEPs), and 5 (241 DEPs). Conversely, *t*-tests often identified very few or no DEPs. Conversely, in W1 (1463 DEPs) and W4 (545 DEPs), the Bayesian method yields a more intermediate count. The *t*-Welch method, along with *t*-Student, appears to be the most conservative in certain workflows, notably identifying very few or no DEPs in W2 (2 DEPs for *t*-Welch, 9 for *t*-Student), W3 (0 DEPs for both), and W5 (7 DEPs for *t*-Welch, 29 for *t*-Student). Limma and DEqMS generally show intermediate numbers, often clustering with *t*-Student and *t*-Welch, though their specific counts vary. It was also observed that the greater the number of identifications per method, the greater the tendency to contribute a larger set of unique DEPs.

UpSet plots (Figure 2B) show that while a substantial core of proteins is shared across methods, each method also contributed unique identifications. Distribution analyses of adjusted *p*-values (Appendix A) confirmed that MSstats tends to produce broader right tails, reflecting higher sensitivity, whereas *t*-tests yield more conservative distributions.

Further analysis of these patterns in relation to the data quality metrics in Table 2 provides additional context. Workflows W2, W3, and W5, where the Bayesian method identified the highest number of DEPs, were characterized by diverse variance patterns. For instance, W2 and W5 exhibited relatively high median CVs for both conditions and non-significant Levene’s test *p*-values (0.9991 for W2; 0.1310 for W5), indicating more homogeneous variance but higher overall variability. In contrast, W3 had very low CVs but a significant Levene’s test *p*-value (0.0221), suggesting heteroscedasticity despite high precision. Workflows W1 and W4, where MSstats identified the most DEPs, showed different characteristics: W1 had low CVs and homogeneous variance (Levene *p* = 0.3694), while W4 showed a high CV for Condition 2 (31.95%) and strongly heterogeneous variance (Levene *p* < 0.0001).

The distribution of protein intensities and statistical metrics across the six statistical methods and five workflows is shown in Appendix A. In all cases, a sharp peak near the origin (low −log_10_(adjusted *p*-value) or low probability) was observed, corresponding to non-significant proteins. The extent of the right tail, representing statistically significant DEPs, varied across methods. MSstats generally displayed broader distributions that extended to higher −log_10_(adjusted *p*-value) values in several workflows (e.g., W1 and W4), consistent with its higher DEP counts. In contrast, Student’s *t*-test and Welch’s *t*-test produced distributions concentrated at lower −log_10_(adjusted *p*-value) values, with a less pronounced or absent right tail in workflows where they identified very few or no DEPs (e.g., W2, W3, and W5). Limma and DEqMS showed intermediate distributions, with right tails more extended than those of the *t*-tests but less than those of MSstats. A “spike” near the significance threshold was observed in some cases, reflecting clustering of proteins close to the cut-off. Variability in distribution shapes across workflows also reflected differences in data processing and normalization steps.

### 2.3. Biological Relevance in Differential Proteomics Analysis

Beyond statistical significance, the biological relevance of identified proteins is para-mount for meaningful interpretation in proteomics studies. This often involves applying additional filters, such as fold change thresholds or probabilistic assessments from Bayesian analyses, to refine the list of potentially interesting proteins. Figure 3 illustrates the overlaps in significant proteins among different frequentist HTMs (Student’s *t*-test, Welch’s *t*-test, Limma, DEqMS, and MSstats) after the application of these bio-logical relevance filters across Works 1, 2, 3, 4, and 5 (Panels A to D). Student’s and Welch’s *t*-tests are not included in this figure due to an insufficient number of significant results after the initial statistical testing and subsequent filtering for biological relevance.

In general, it is observed that following frequentist hypothesis testing, the filtering for determining biological relevance is more stringent when employing Bayesian methods compared to a simple fold change criterion. This is evidenced by the lower total number of “retained” proteins (Figure 3) and, consequently, a higher number of discarded proteins (Appendix A). Consequently, the selection of both the hypothesis testing method and the biological relevance criterion jointly determines the final set of relevant proteins, directly influencing downstream pathway and functional enrichment analyses.

### 2.4. Functional Enrichment Analysis: Consistency Across Methodological Choices

Following the application of biological relevance filters to identify differentially expressed proteins, functional enrichment analyses were performed. Significant protein lists were categorized as overrepresented or underrepresented based on the developed methodology. Appendix A summarizes the interactions between these methods for both overrepresented and underrepresented proteins. In general, comparisons between protein sets filtered by the Bayesian approach versus fold change showed no substantial differences in either the number of proteins identified or their absolute log_2_FC values. Notable exceptions were observed in Work 1 for underrepresented proteins and in Work 4 for overrepresented proteins. In Work 1, the msstats_FC_up method (MSstats as method for hypothesis testing, fold change as filter to determine biological relevance and overrepresented category) exhibited 1190 interactions, whereas the corresponding Up Bayesian method showed only 82 interactions. Similarly, in Work 4, msstats_FC displayed 489 self-interactions in the Up FC category, compared to 73 for the Up Bayesian category.

To further assess the consistency of the functional enrichment results, we analyzed the distributions of Euclidean similarity and Jaccard index for each individual ‘Work’ and direction of regulation (Figure 4). These metrics were chosen for their distinct yet complementary strengths in assessing similarity between sets of enriched terms. The Jaccard index, a binary metric, quantifies the overlap based solely on the presence or absence of shared terms, providing a direct measure of commonality regardless of their quantitative enrichment levels. In contrast, Euclidean similarity, a distance-based metric, considers the quantitative relationships of terms (e.g., based on the % Associated Genes scores from ClueGO), making it sensitive to both shared terms and the magnitude of their enrichment. Together, these metrics offer a robust framework to evaluate consistency, capturing both qualitative (Jaccard) and quantitative (Euclidean) aspects of functional enrichment profiles. Across all five Works (W1–W5) and for both overrepresented and underrepresented proteins, comparisons in which the Hypothesis Testing Method (HTM) varied while the Criterion for Biological Relevance (CBR) remained constant (Intra-HTM_FC_CBR and Intra-HTM_Bayes_CBR categories) exhibited consistently high similarity values (Figure 4). In contrast, comparisons in which the CBR varied while the HTM was held constant (Intra-CBR_Fixed_HTM) showed lower similarity values across most Works and directions. The lowest similarity values were observed in Inter-HTM/Inter-CBR comparisons, where both HTM and CBR differed. In contrast, comparisons where the CBR was varied while the HTM was kept constant (Intra-CBR_Fixed_HTM category) consistently showed lower similarity values for both Euclidean similarity and Jaccard index across most ‘Works’ and directions (Figure 4).

A comprehensive summary of identified proteins, including median, minimum, and maximum log_2_FC values for each HTM and CBR across all Works, is provided in Appendix A. Additional analyses for Work 1 are shown in Appendix A, including boxplots of Pearson and Spearman correlation coefficients and heatmaps of Euclidean, Jaccard, Pearson, and Spearman metrics across the three core ontologies (GO Biological Process, GO Molecular Function, and KEGG), with separate panels for overrepresented and underrepresented proteins. These analyses captured both the consistency observed when the CBR is held constant and the variability arising from changes in CBR, as exemplified by W1 and W4. Collectively, Appendix A and Figure 4 and Appendix A provided detailed quantitative results on the consistency of functional enrichment outcomes across methodological choices.

### 2.5. Meta-Analysis of Functional Enrichment Concordance

Finally, a meta-analysis was performed with the objective of systematically evaluating the impact of diverse statistical methodologies on the outcomes of biological enrichment analysis. This meta-analysis compiled Euclidean similarity and Jaccard indices from multiple quantitative proteomics “Works”, categorizing them into four primary comparison types based on the interplay between the HTM and the CBR: “Intra-HTM_FC_CBR”, “Intra-CBR_Fixed_HTM”, “Inter-HTM_Inter-CBR”, and “Intra-HTM_Bayes_CBR”. An “Unknown” category with very few data points was also observed, representing method combinations not fitting the primary classifications; due to its sparse representation, the interpretation focused on the four main comparison types. The distribution of Euclidean similarity and Jaccard indices, both in their original and arcsin square root-transformed forms, are presented in Figure 5, Panels A and C (histograms), respectively. The arcsin square root transformation generally shifts the distribution towards a more symmetrical form, which is beneficial for statistical analyses. Global distribution and comparison of Pearson and Spearman indices from meta-analysis are presented in Appendix A.

The results are presented both globally (across all analyzed ontologies and Works combined in Figure 5A,C—boxplots) and specifically for key Gene Ontology (GO) and KEGG pathway ontologies (Figure 5B,D). The global analysis of Euclidean similarity (Figure 4, left panels for each category boxplot) revealed a highly significant overall difference in similarity distributions among the comparison types (Kruskal–Wallis *p* = 0.00011). For Jaccard index (Figure 5C, boxplot), a similar highly significant overall difference was observed (Kruskal–Wallis *p* < 2 × 10^−16^). When comparing the median Euclidean similarities and Jaccard indices across the categories, it was observed that “Intra-HTM_FC_CBR” and “Intra-HTM_Bayes_CBR” showed the highest median similarity, “Intra-CBR_Fixed_HTM” showed a lower value and “Inter-HTM_Inter-CBR” the lowest levels of consistency.

Ontology-specific analysis (Figure 5B for Euclidean and 5D for Jaccard) largely maintains these global trends across ontologies. For instance, in GO Biological Process and GO Molecular Function, “Intra-HTM_FC_CBR” and “Intra-HTM_Bayes_CBR” generally exhibited higher overlap, while “Intra-CBR_Fixed_HTM” and “Inter-HTM_Inter-CBR” consistently showed lower consistency.

Finally, a sensitivity analysis was performed by systematically excluding one “Work” (dataset) at a time and re-running the Kruskal–Wallis test on the remaining data (Figure 5B,D, bottom bar plots). For Euclidean similarity, excluding Work 1, Work 3, or Work 4 resulted in the Kruskal–Wallis *p*-value for global differences increasing above the 0.05 significance threshold (e.g., *p* = 0.22 for W1 excluded in Figure 5B). Similarly, for Jaccard index, excluding Work 1 also caused the *p*-value to exceed 0.05.

## 3. Discussion

This study presents a novel meta-analytical approach by combining enrichment results from diverse real-world, independently published quantitative proteomics datasets rather than controlled benchmark datasets. This allows for a comprehensive evaluation of the relative influence of both specific HTMs and distinct CBRs on downstream biological interpretations, reflecting the variability encountered in actual research. While method benchmarking studies often utilize specially prepared datasets to validate new approaches, the use of a meta-analysis on randomly selected or pre-existing “real” datasets is less common and offers valuable insights into the generalizability and robustness of analytical choices in routine proteomics research. By analyzing the different Works using parameters distinct from those in the original studies (which could explain possible differences with respect to them), we aim to address three key questions regarding the influence of statistical methodologies on biological enrichment findings:Does the specific hypothesis testing method (HTM; e.g., *t*-Student, *t*-Welch, Limma, DEqMS, MSstats, Bayesian) influence the resulting biological enrichments when the criterion for biological relevance (CBR) is kept constant?Does the method used for determining biological relevance (CBR; fold change-based vs. Bayesian posterior probability-based approaches) influence the resulting biological enrichments when the hypothesis testing method (HTM) is kept constant?What has a greater influence on the observed biological enrichments: the specific hypothesis testing method (HTM) or the criterion for determining biological relevance (CBR)?

### 3.1. Methodological Implications of Hypothesis Testing in Proteomics

Our comparative framework demonstrates that a key factor shaping differences in the number of differentially expressed proteins identified by HTM application is the inherent variability in protein quantification within each dataset. The analysis of median coefficients of variation (CVs) across conditions (Table 2, Appendix A) reveals substantial differences among datasets. Works 4 and 5 display pronounced disparities between conditions: in Work 4, Cond2 shows a higher median CV (31.95%) compared to Cond1 (18.62%), while in Work 5, Cond1 exhibits a higher median CV (36.42%) than Cond2 (31.99%). Notably, for these datasets, Cond2 was derived from the same raw files, indicating that these differences primarily reflect the impact of normalization strategies applied across replicates rather than sample composition. High CVs may indicate increased biological heterogeneity, technical variability, or a combination thereof, and are likely to influence both the identification and reproducibility of DEPs. Complementary analyses using Levene’s test further highlight that certain datasets may require HTMs capable of handling unequal variances.

Similarly, the choice of hypothesis testing method (HTM) markedly influences the number and composition of differentially expressed proteins (DEPs). Methods such as MSstats and Bayesian modeling tended to produce higher DEP counts, whereas classical approaches, including Student’s and Welch’s *t*-tests, were more conservative, occasionally identifying few or no significant proteins (Figure 2). This variability highlights the dependency of DEP lists on the chosen statistical approach, a well-documented challenge in proteomic data analysis. Different statistical models emphasize distinct aspects of the data, potentially due to variations in assumptions regarding variance estimations or outlier handling. When MSstats and the Bayesian method identified the highest DEP counts, this may reflect greater sensitivity, robustness and adaptability to common proteomics challenges like missing values and outliers. For instance, MSstats is designed to manage missingness through imputation or probabilistic modeling, while Limma’s empirical Bayes moderation stabilizes variance estimates, particularly with small sample sizes. When these methods are applied to data where such issues are effectively managed, their convergence on similar significant findings underscores their reliability, allowing them to capture a broader range of subtle changes. Conversely, this approach may also yield a higher proportion of unique findings that are potentially less robust.

Overall, these observations indicate that both the choice of HTM and dataset-specific characteristics—including within-condition variability, fold change dispersion, and variance equality—affect the reproducibility and robustness of differential expression results. Robust methods such as MSstats or Bayesian approaches may be particularly beneficial for datasets with high CVs or unequal variances, ensuring reliable identification of DEPs and appropriate contextualization of downstream functional enrichment analyses.

### 3.2. The Criterion for Biological Relevance as the Key Driver

In contrast, the choice of CBR—fold change versus Bayesian posterior probability—emerges as the most decisive factor shaping downstream interpretations. Our analyses show a clear reduction in concordance when CBRs are varied under a fixed HTM, whereas different HTMs under the same CBR yield consistent enrichment profiles (Figure 4, Figure 5, Appendix A). This observation is consistent with previously noted discrepancies in the number of identified proteins for certain Works when comparing Bayesian and fold change filtering. For instance, the ‘Underrepresented (down)’ panel for W1 in Figure 4 visually confirms lower similarity values in the Intra-CBR_Fixed_HTM category, consistent with the noted difference in protein counts for underrepresented proteins in W1. Similarly, while not directly shown for W4’s overrepresented proteins at this specific granularity, the general trend supports that changes in CBR lead to greater divergence. Additionally, in the meta-analysis, comparisons of Euclidean similarities and Jaccard indices within the Intra-HTM_FC_CBR category reveal a consistently high degree of agreement among different frequentist HTMs when identifying enriched terms using a fold change (FC)-based criterion for biological relevance. This indicates that, within the frequentist framework, the specific choice of HTM has a relatively minor impact on the resulting biological enrichments. This pattern is also observed for the Intra-HTM_Bayes_CBR category since its consistency remains high when varying HTMs solely within the Bayesian framework (Figure 5 and Appendix A). Indeed, Intra-CBR_Fixed_HTM showed a profound drop in consistency, highlighting that changing the criterion for biological relevance (from FC to Bayesian) while keeping the HTM fixed leads to substantial divergence in functional enrichment results.

Bayesian filtering, although more conservative, provides a probabilistic and data-driven measure of biological relevance, limiting the inclusion of spurious findings that may pass arbitrary fold change thresholds. At the same time, its stringency can exclude pathways that appear under fold change filtering (e.g., Parkinson’s disease with DEqMS; ER protein processing with MSstats). This highlights a methodological trade-off between inclusiveness and robustness, reinforcing that the definition of biological relevance is more critical than the choice of the statistical test itself.

### 3.3. Reproducibility, Transparency, and Community Standards

The discrepancies observed between our repeated analyses and the results originally reported by dataset authors underscore the influence of differences in normalization, search parameters, and replicate handling. Such variability illustrates the necessity of standardized protocols, careful batch-effect management, and defined quality-control checkpoints to reduce technical noise. Equally important is transparent reporting: adherence to FAIR principles and established proteomics standards (MIAPE, HUPO-PSI) will enable comparability, method benchmarking, and data reuse. Providing raw inputs to enrichment tools (protein IDs, ClueGO outputs, parameters), code, and environment specifications (R package versions, seeds) minimizes ambiguity and allows direct replication by other groups.

As concrete examples of divergence, in Work 1 (up-regulated proteins) we observed that some pathways were consistently detected across both filtering strategies. For instance, the KEGG pathway “Complement and coagulation cascades” (hsa04610) and the GO term “acute-phase response” (GO:0006953) were significant under both Bayesian- and FC-based criteria when using DEqMS and MSstats. However, many other terms appeared exclusively in the FC-filtered lists. With MSstats, pathways such as “Protein processing in endoplasmic reticulum” (hsa04141) and “translation” (GO:0006412) were enriched only under FC filtering, whereas the Bayesian approach did not retain them. Similarly, with DEqMS we found disease-related pathways, including “Parkinson’s disease” (hsa05012), to be highlighted only under FC filtering. Notably, no terms were found exclusively after these HTMs with the Bayesian criterion, suggesting that this approach was more conservative. These examples illustrate how the choice of biological relevance criterion can markedly alter the biological interpretation, either by converging on robust terms or by excluding potentially relevant pathways depending on the filtering strategy.

### 3.4. Limitations and Future Perspectives

Our sensitivity analyses reveal that individual datasets can substantially affect global statistical significance, indicating that conclusions drawn from meta-analyses are dataset-dependent. Future studies using more homogeneous biological systems will be needed to test whether the observed patterns generalize across contexts. Furthermore, proteomics results require orthogonal validation—Western blot, ELISA, targeted MS (PRM/SRM)—to confirm protein-level findings and functional assays to contextualize biological significance [28]. Another challenge is the dominance of highly abundant proteins, which can obscure subtler but more informative patterns. Component-based strategies that disentangle “size” and “shape” effects, as proposed by Roden [29], offer a promising complementary approach to enrichment reproducibility analyses and should be explored in future work.

## 4. Materials and Methods

### 4.1. Dataset Selection

Five publicly available mass spectrometry (MS) proteomic works (referred to as Works 1–5) were selected for this study. Works 1, 2, and 3 were chosen randomly based on three criteria: their acquisition using an Orbitrap Fusion mass spectrometer, utilization of data-dependent acquisition (DDA) mode, and prior publication in peer-reviewed journals. Works 4 and 5 were selected from a previously published work by our research group, adhering to the same criteria. From each work, specific RAW files were obtained to perform pairwise comparisons. Information for each work is detailed below:-Work 1 (ProteomeXchange: PXD051640) originated from a study on brown adipose tissue and liver in a cold-exposed cardiometabolic mouse model [30]. The protein database used for identification was *Mus musculus* (C57BL/6J) (UP000000589).-Work 2 (ProteomeXchange: PXD041209) investigated the *Escherichia coli* protein acetylome under three growth conditions [31]. Protein identification relied on the *Escherichia coli* K12 (UP000000625) protein database.-Work 3 (ProteomeXchange: PXD019139) explored quantitative proteome and PTMome responses in *Arabidopsis thaliana* roots to osmotic and salinity stress [32]. The corresponding protein database was Arabidopsis thaliana (UP000006548).-Works 4 and 5 (ProteomeXchange: PXD034112) were derived from a comprehensive study on biological nitrogen fixation and phosphorus mobilization in *Azotobacter chroococcum* NCIMB 8003 [33]. For dataset 4, raw files from control and biological nitrogen fixation conditions were used, while for dataset 5, raw files corresponding to control conditions were compared to phosphorous mobilization conditions. The protein database for these works was UP000068210”.

All files were analyzed in MaxQuant, using the parameters specified in Appendix A.

### 4.2. Differential Abundance Analysis

Data processing and statistical analyses were primarily conducted using R (version 4.5.0) [34] within the RStudio 2025.05.0 environment, leveraging various specialized packages. The initial input for differential abundance analysis consisted of the proteinGroups.txt and evidence.txt files, which are standard outputs from the MaxQuant processing. As a crucial data preparation step (Script 1), contaminants, proteins identified by only one unique peptide, and proteins from the decoy database were filtered out from the proteinGroups.txt file and column names were standardized for consistent downstream processing. The resulting filtered dataset, named proteinGroups_filtered.txt, was then used as the primary input for most downstream statistical analyses. For all analyses, LFQ normalized intensities data were utilized. Missing values were not globally imputed. Instead, missingness was handled method-specifically, in line with each tool’s underlying assumptions. The selected statistical methods were robust to incomplete data, allowing valid inference without requiring full imputation. Six distinct hypothesis testing methods (HTMs) were applied to these prepared datasets (Script 2):-Student’s and Welch’s *t*-tests: Performed on the base-2 logarithm of protein intensity data to compare means between two conditions. Both Student’s *t*-test (assuming equal variances) and Welch’s *t*-test (not assuming equal variances) were applied using pairwise complete observations; that is, proteins were retained if they had at least two valid (non-missing) values per group.-Limma: The Limma R package [17] was used to fit a linear model to log2-transformed protein intensity data. This method employs empirical Bayes moderation of variances and tolerates missing values, provided sufficient replicate data are available. In this way, statistical power in enhanced and variance stabilized, which is particularly critical in experiments with low biological replicates.-DEqMS: The DEqMS R package [18] was employed, extending the Limma framework by incorporating peptide count information to refine variance estimation in differential protein abundance analysis. It leverages the observation that proteins identified with more peptide-spectrum matches (PSMs) yield more reliable intensity measurements, leading to improved statistical power. As with Limma, DEqMS accepts missing values natively during model fitting and uses empirical variance estimation without requiring imputation.-MSstats: The MSstats R package [13] is specifically designed for quantitative mass spectrometry data. Uniquely among these methods, MSstats requires the original proteinGroups.txt file (not the filtered version) along with the evidence.txt file as input. Data was pre-processed using the dataProcess function in MSstats, and group comparisons were performed using linear mixed-effects models. This approach accounts for various sources of variability (e.g., biological/technical replicates, batch effects) by explicitly modeling them as random effects, thus providing robust variance estimates and increased statistical power. Missing values were handled using MSstats’ default model-based approach, which treats them as censored (i.e., non-random) and does not perform global imputation.-Bayesian Analysis: Differential protein abundance was also assessed using a Bayesian framework, implemented with the rstanarm R package [25]. This package provides an interface to Stan for Hamiltonian Monte Carlo (HMC) sampling. For each protein, a Bayesian linear regression model was fitted to the log2-transformed intensity data, utilizing the experimental condition as predictor. Only proteins with at least two valid values per group were considered. Missing values were implicitly handled through marginalization over the posterior, without requiring imputation. Model fitting employed four Markov Chain Monte Carlo (MCMC) chains with 4000 iterations (including 2000 warm-up iterations) and an adapt_delta of 0.99. Convergence of the chains was rigorously monitored using Rhat values (ideally ≤1.01) and effective sample size (ESS, ideally ≥200). This probabilistic approach yields full posterior distributions for the model parameters, which enables direct statements about effect sizes and their associated uncertainties. Weakly informative priors were incorporated to regularize parameter estimates and enhance model stability, particularly beneficial for proteins with limited measurements [35].

For all frequentist methods (Student’s *t*-test, Welch’s *t*-test, Limma, and DEqMS), *p*-values were adjusted for multiple testing using the Benjamini–Hochberg (BH) method to control the false discovery rate. Proteins with an adjusted *p*-value ≤ 0.05 were considered significantly differentially abundant. For the Bayesian method, the posterior probability of biological relevance was calculated, representing the likelihood that the absolute effect size (log2FC) exceeded a predefined threshold (e.g., 1). Proteins with a posterior probability ≥ 0.95 were then considered differentially abundant and biologically relevant.

To provide a comprehensive overview of the differential abundance analysis results, several types of plots were generated for each pairwise comparison (Script 2): bar plots (using the ggplot2 package) visualizing the total number of significant proteins identified by each method; UpSet plots (using the UpSetR package, [36]) representing the intersections and unique sets of significant proteins across different methods, complemented by tabular summaries of these intersections; and density plots of −log_10_(adjusted *p*-value) (using ggplot2) to assess overall trends in *p*-value distributions from frequentist methods, with non-finite or zero *p*-values excluded from the transformation for accurate representation, including a vertical line for the significance cutoff.

### 4.3. Biological Relevance Filtering and Overlap Analysis

To identify proteins with significant biological relevance beyond mere statistical significance, two distinct criteria for biological relevance (CBRs) were applied to the initial results of differential abundance analysis (Script 3). The first criterion, fold change (FC) filtering, was applied to proteins statistically identified as differentially abundant by the frequentist methods (*t*-Student, *t*-Welch, Limma, DEqMS, MSstats). A protein was considered biologically relevant if its absolute log2 fold change (∣log2FC∣) was greater than or equal to 1. The second criterion, Bayesian Biological Relevance Filtering, employed the previously described Bayesian linear modeling approach. After the initial hypothesis testing, for each protein, a Bayesian linear model (fitted using the rstanarm R package [25] with the MCMC parameters and convergence diagnostics as detailed above) was utilized. A protein was deemed biologically relevant by this criterion as previously described.

To understand the agreement and unique contributions of each filtering strategy, intersection analysis was performed using UpSet plots [36]. For each statistical method, three sets of proteins were defined: “Originals” (statistically significant proteins from the HTMs), “FC” (proteins from the “Originals” set also meeting the fold change CBR), and “Bayes” (proteins from the “Originals” set also meeting the Bayesian CBR). These sets of protein identifiers were used as input for the UpSetR package in R. UpSet plots were generated to visualize the size of unique sets and all possible intersections.

### 4.4. Segregation and Functional Enrichment Analysis

Following the HTM analysis and CBR filtering, proteins were classified as up- or down-regulated based on their log2 FC values and the respective statistical or biological relevance thresholds (Script 4). For each HTM and CBR, separate lists of up-regulated and down-regulated protein identifications (Protein.IDs) were generated and Over-Representation Analysis (ORA) was developed. Importantly, while the res_bayes method originates from Bayesian inference, its results for biological relevance were also considered under the fold change criterion for specific downstream applications.

Functional enrichment analysis was performed using ClueGO (v2.5.9) [37] within Cytoscape (v3.10.0) [38]. For each set of up- and down-regulated proteins, GO (Biological Process and Molecular Function) terms and KEGG pathways were deliberately interrogated to maintain a standardized and comparable analysis framework across species. The enrichment analysis relied on a two-sided hypergeometric test, with resulting *p*-values corrected for multiple testing using the Benjamini–Hochberg method. Only terms with a corrected *p*-value < 0.05 were considered significant. To reduce redundancy and improve interpretability, functionally related terms were grouped based on their kappa score using the GO Term Fusion option, and the resulting networks were visualized based on the overlap of associated genes.

### 4.5. Similarity and Correlation Analysis of Functional Enrichment Outcomes

To systematically evaluate the impact of Hypothesis Testing Methods (HTMs) and Criteria for Biological Relevance (CBRs), we extracted and analyzed the % Associated Genes data from each dataset, for both upregulated and downregulated genes. This was performed for each HTM and CBR framework within individual quantitative proteomics datasets. This quantitative information, representing the strength of enrichment for each term, allowed for a comprehensive assessment of agreement using various metrics (Script 5). The following similarity and correlation metrics were calculated:-Jaccard Index (J(A,B) = ∣A∪B∣∣A∩B∣): This metric was used to quantify the overlap between the sets of enriched terms (defined as terms with % Associated Genes > 0) derived from different method combinations. For statistical analysis, Jaccard Index values were transformed using the arcsin square root transformation (arcsin(x)) to stabilize variance.-Pearson Correlation Coefficient: This metric assessed the linear relationship between the quantitative profiles (vectors of % Associated Genes for all terms within a given ontology) generated by different method combinations.-Spearman Correlation Coefficient: This non-parametric metric evaluated the monotonic relationship between the quantitative profiles of % Associated Genes, making it robust to non-linear associations and outliers.-Euclidean Similarity: Derived from Euclidean distance, this metric quantified the closeness of the quantitative profiles of % Associated Genes between different method combinations. Similarity was calculated as 1/(1 + d), where d is the Euclidean distance, resulting in values ranging from 0 (maximal dissimilarity) to 1 (maximal similarity).

These similarity and correlation indices were rigorously categorized based on the nature of the combined HTM and CBR for pairwise comparisons, mirroring the scheme implemented in our analysis scripts:-Intra-HTM_FC_CBR: Comparisons between different HTMs where the CBR was consistently fold change-based (e.g., deqms_FC vs. Limma_FC). This category assesses the variability introduced solely by the choice of HTM when a fixed FC relevance criterion is applied.-Intra-HTM_Bayes_CBR: Comparisons between different HTMs where the CBR was consistently Bayesian posterior probability-based (e.g., deqms_Bayes vs. Limma_Bayes). This category assesses the variability introduced solely by the choice of HTM when a Bayesian relevance criterion is applied.-Intra-CBR_Fixed_HTM: Comparisons between the two different CBRs (fold change-based vs. Bayesian posterior probability-based), where the HTM was kept constant (e.g., tstudent_FC vs. tstudent_Bayes). This category directly evaluates the influence of the biological relevance criterion itself, controlling for the HTM.-Inter-HTM_Inter-CBR: Comparisons between combinations where both the HTM and the CBR differed (e.g., tstudent_FC vs. Limma_Bayes). This category represents the cumulative variability from changing both methodological aspects.

For each individual “Work” and for each direction of regulation (up/down), a non-parametric Kruskal–Wallis H-test (*p* < 0.05) was performed to assess overall differences in the distributions of each metric (Jaccard, Pearson, Spearman, Euclidean Similarity) across these defined comparison types. In cases where the test could not be reliably computed (e.g., due to insufficient data variability resulting in an NA *p*-value), the test was skipped, and this was noted in the analysis logs. If significance was detected, post hoc Dunn’s tests with Bonferroni correction [39] were performed to identify specific pairs of groups with significantly different distributions.

Furthermore, heatmaps were generated for each similarity and correlation matrix (Jaccard Index, Pearson Correlation, Spearman Correlation, and Euclidean Similarity) for each Work, direction, and ontology, providing a visual representation of agreement patterns. Boxplots illustrating the distribution of each metric across the defined comparison types were also generated for each Work and direction, as well as a consolidated global analysis across all Works and directions.

### 4.6. Meta-Analysis

A comprehensive meta-analysis was performed to evaluate the consistency of biological enrichment results across various quantitative proteomics datasets and statistical methodologies. The process began with the consolidation of individual ClueGO enrichment outputs (originally in .xls format) into structured tab-separated value (TSV) files for each “Work” and direction of regulation (up/down). During this consolidation, for terms enriched by a given method, the mean of the “Associated Genes” percentage was calculated if multiple entries for the same term and ontology existed, and missing values (NA) for non-enriched terms were imputed as zero. This refined data structure, which included “Term” and “Ontology” alongside the “% Associated Genes” for each applied method, served as the foundational input for the meta-analysis (Script 6).

Beyond Jaccard similarity, the consistency between all pairwise combinations of statistical methods and biological relevance criteria was quantified using three additional metrics: Pearson correlation, Spearman correlation, and Euclidean similarity. While Jaccard index assessed the overlap of enriched terms (terms with “% Associated Genes” > 0), Pearson and Spearman correlations evaluated the linear and monotonic relationships, respectively, of the “% Associated Genes” values themselves. Euclidean similarity provided a metric of overall proximity between the quantitative profiles.

Raw ontology names were systematically standardized to their core functional categories (e.g., “GO_BiologicalProcess”, “GO_MolecularFunction”, “KEGG”) to facilitate cross-Work comparisons. All extracted Jaccard indices were then transformed using the arcsin square root transformation (arcsin(√J)) to improve normality and homogeneity of variance, a common practice for proportional data. Pearson, Spearman, and Euclidean similarity values were used directly for analysis as their ranges ([−1, 1] for correlations and [0, 1] for Euclidean similarity) were already suitable for statistical interpretation. These transformed (or direct) indices were combined into a single, comprehensive dataset, along with metadata detailing the original Work, normalized ontology, direction of regulation (up/down), and their specific methodological comparison category. Four distinct comparison categories were established: Intra-HTM_FC_CBR (different Hypothesis Testing Methods (HTMs) with consistent fold change-based Criteria for Biological Relevance (CBR)), Intra-HTM_Bayes_CBR (different HTMs with consistent Bayesian posterior probability-based CBR), Intra-CBR_Fixed_HTM (different CBRs with a fixed HTM), and Inter-HTM_Inter-CBR (both HTM and CBR differed).

Statistical analysis for the meta-analysis was conducted using the non-parametric Kruskal–Wallis test to assess overall differences in metric distributions across these comparison types. This was followed by Dunn’s post hoc test with Bonferroni correction for pairwise comparisons when global significance was observed. The robustness and consistency of the overall findings were further evaluated through a sensitivity analysis, where the Kruskal–Wallis test was re-run by systematically excluding one ‘Work’ at a time from the meta-analysis dataset.

All analyses and visualizations were performed using R (version 4.5.0), leveraging the tidyverse suite for data manipulation, ggpubr for statistical tests and visualization, patchwork for combining plots, readxl for .xls file input, and dunn.test for post hoc analysis. Scripts are available in https://github.com/AlfonsoOA/HTM-CRB and on Zenodo (DOI: 10.5281/zenodo.16541357), accessed on 28 July 2025.

## 5. Conclusions

This meta-analysis underscores the pivotal role of methodological choices in shaping the outcomes of quantitative proteomics studies. By systematically comparing HTMs and CBRs across five independent datasets, we demonstrate that the definition of biological relevance exerts a far greater influence on downstream functional enrichment profiles than the choice of hypothesis testing method.

The Bayesian per-protein approach provides a more rigorous and probabilistic framework than arbitrary fold change cutoffs, offering greater methodological defensibility and biological accuracy, albeit with more conservative results. Conversely, fold change cutoffs can yield broader inclusiveness but risk inflating false positives. Together, these findings emphasize the necessity of transparent reporting and adherence to community standards (FAIR, MIAPE, HUPO-PSI) to ensure reproducibility and comparability across proteomics research.

Ultimately, reliable biological interpretation requires both methodological rigor and independent validation. By integrating robust statistical criteria, standardized workflows, and orthogonal validation strategies, proteomics can provide more reproducible and biologically meaningful insights into complex systems.

## Figures and Tables

**Figure 1 ijms-26-09232-f001:**
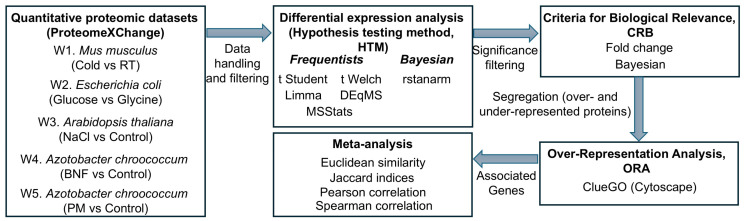
General workflow of the analytical pipeline. The schematic diagram illustrates the step-by-step methodology for the comparative analysis. Quantitative proteomic datasets from five different studies (W1-W5) were subjected to differential expression analysis using six distinct Hypothesis Testing Methods (HTMs). The results were then filtered using Criteria for Biological Relevance (CBR) and segregated into lists of upregulated and downregulated proteins. Over-Representation Analysis (ORA) was performed on these lists to identify functional categories, followed by a meta-analysis using various similarity and correlation metrics to assess the impact of methodological choices.

**Figure 2 ijms-26-09232-f002:**
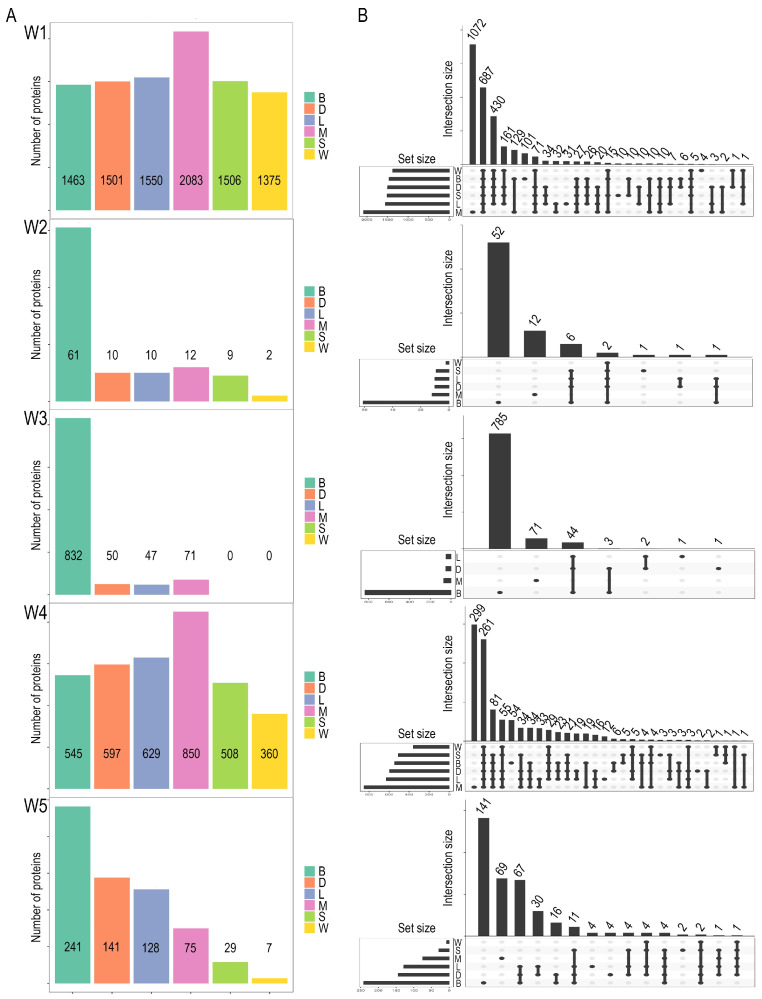
Number and intersection of differentially abundant proteins across Hypothesis Testing Methods (HTMs) for five quantitative proteomics Works. (**A**) Bar plots show the total number of significant proteins identified by each method: Bayesian (B), DEqMS (D), Limma (L), MSstats (M), Student’s *t*-test (S), and Welch’s *t*-test (W). (**B**) UpSet plots show the intersections of significant protein sets across methods for each Work. Vertical bars indicate the size of each intersection; connected dots below the bars identify which methods contribute to that intersection. Shared intersections highlight proteins consistently identified as significant across methods, while unique intersections reflect method-specific findings.

**Figure 3 ijms-26-09232-f003:**
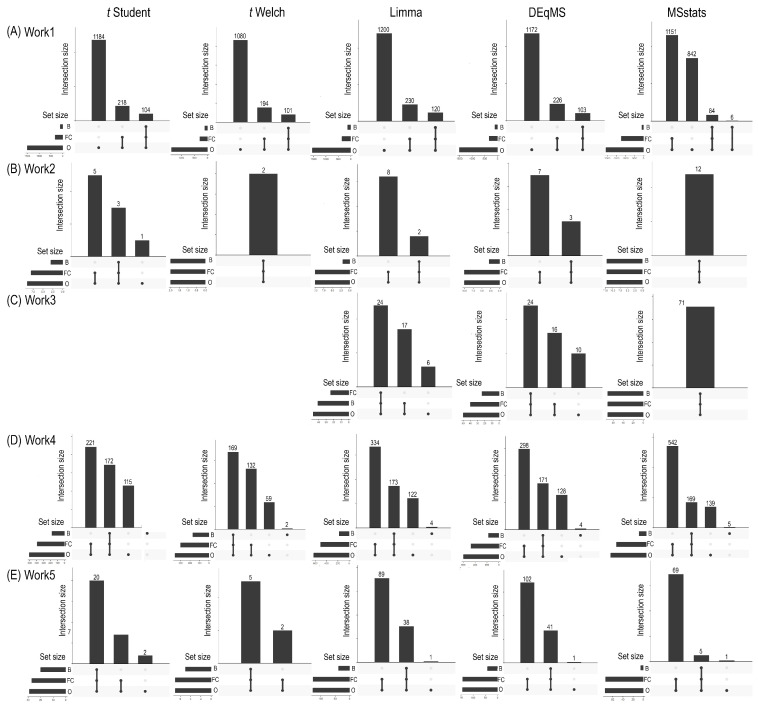
Comparison of overlaps in the significant proteins in the different hypothesis tests after the biological relevance filter (fold change and Bayesian analysis). The overlap results of the identified proteins are presented for each of the frequentist methods (Student’s *t*, Welch’s *t*, Limma, DEqMS and MSstats) in Works 1, 2, 4 and 5 (**A**–**E**). Work 3 did not yield sufficiently significant results. B, proteins identified only after applying the biological relevance filter by the Bayesian method; FC, proteins identified only after applying the fold change filter; O, all proteins present after the hypothesis test.

**Figure 4 ijms-26-09232-f004:**
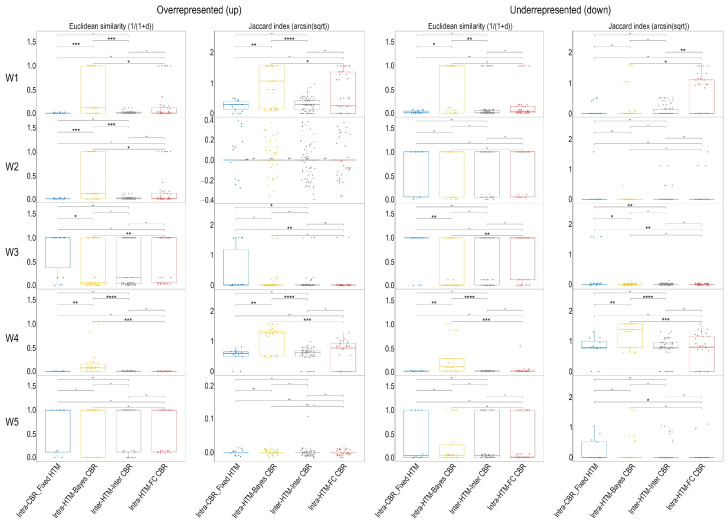
Euclidean similarity and Jaccard index distributions for each Work and regulation direction. Each panel contains boxplots of the metric for four comparison categories (Intra-HTM_FC_CBR, Intra-HTM_Bayes_CBR, Intra-CBR_Fixed_HTM, Inter-HTM_Inter-CBR). Boxes = IQR; whiskers = 1.5 × IQR; points = individual pairwise comparisons. Asterisks denote pairwise significance after Dunn’s test with Bonferroni correction. Note: Jaccard is computed on presence/absence of enriched terms (%AssociatedGenes > 0), Euclidean similarity is 1/(1 + d) with d the Euclidean distance on %AssociatedGenes. See Methods for details. ns: not significant, * *p* < 0.05, ** *p* < 0.01, *** *p* < 0.001, **** *p* < 0.0001.

**Figure 5 ijms-26-09232-f005:**
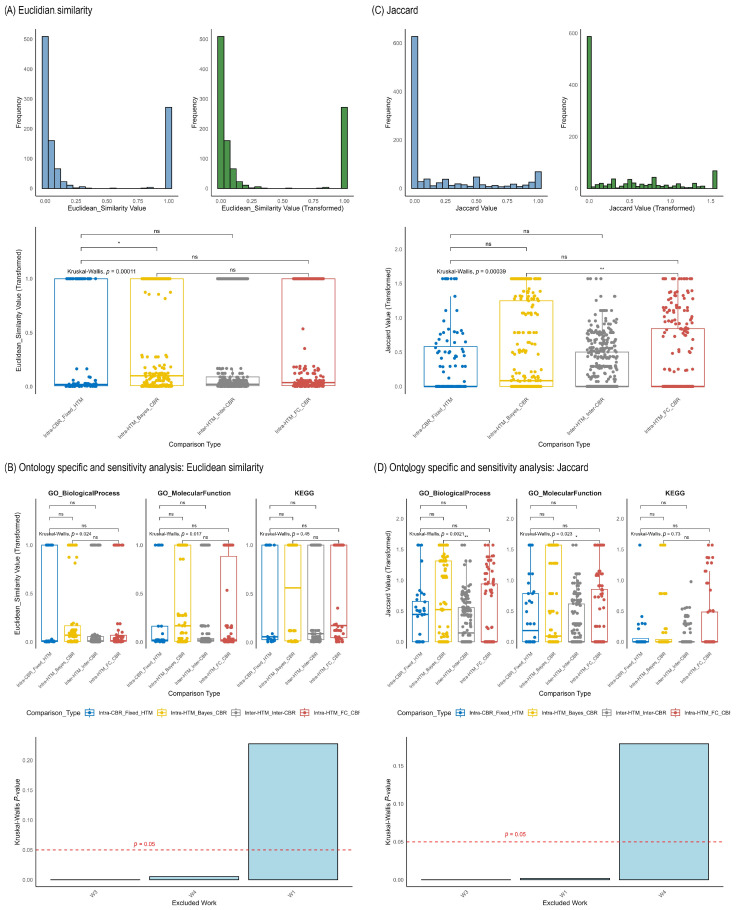
Global distribution and comparison of Euclidean similarity and Jaccard indices from meta-analysis. (**A**,**C**) Histogram showing the distribution of the raw Euclidean similarity and Jaccard Index values, respectively, across all collected data (**left**), histogram of the arcsin(sqrt)-transformed values, illustrating how the transformation affects the data distribution (**right**) and boxplots representing the arcsin(sqrt)-transformed Euclidean similarity and Jaccard Index values, categorized by four comparison types: “Intra-CBR_Fixed_HTM” (concordance among CBRs with fixed HTM), “Intra-HTM_Bayes_CBR” (concordance among Hypothesis Testing Methods with Bayesian-based Criteria for Biological Relevance), “Inter-HTM_Inter-CBR” (concordance when both HTM and CBR vary), “Intra-HTM_FC_CBR” (concordance among HTMs with FC-based CBR) (down). Asterisks denote significance from post hoc Wilcoxon rank-sum tests (ns: not significant, * *p* < 0.05, ** *p* < 0.01). (**B**,**D**) Sensitivity Analysis of Kruskal–Wallis Test for Euclidean similarity and Jaccard Index. Bar plot showing the Kruskal–Wallis *p*-value for the global comparison when sequentially excluding one ‘Work’ (dataset) at a time from the meta-analysis. The red dashed line at *p* = 0.05 serves as a significance threshold.

**Table 1 ijms-26-09232-t001:** Recommended statistical tests for quantitative proteomics differential expression analysis based on experimental design.

Experimental Design	Most Commonly Used Test	Other Possible Test
Simple Comparison (A vs. B)	Student’s *t*-test [16]	Limma (moderated *t*-test) [17], DEqMS [18], Bayesian models [19]
Multiple Conditions	One-way ANOVA [16]	Limma [16], DEqMS [18], Bayesian models [19]
Time Series Experiments	ANOVA/Linear Regression [16]	Linear mixed-effects models (MSstats) [19], Limma [17], DEqMS [18], Bayesian [19]
Multifactorial (e.g., treatment × time)	Factorial ANOVA [16]	Mixed-effects models (MSstats) [13], Limma [17], DEqMS [18], Bayesian [19]
Controlled Reference Mixtures	ANOVA/*t*-test [16]	Limma [17], DEqMS [18], Bayesian [19]
Spectral Count Data	QSpec [14]	QSpec [14], hierarchical Bayesian count models [19]
Extended Time Series (>4 points)	Regression/Clustering [16]	Linear mixed-effects models (MSstats) [13], Bayesian time series [19]
Low Replication Designs	*t*-test/PLGEM-STN [20]	PLGEM-STN [20], Limma [17], DEqMS [18], Bayesian [19]

**Table 2 ijms-26-09232-t002:** Protein groups variability. Cond1, condition 1 (Cold, Glu, NaCl, BNF and PM, respectively, for each Work); Cond2, condition 2 (RT, Gly, No saline stress, no BNF, no PM, respectively, for each Work). Levene *p* value shows the result of the Levene’s test (equality of variances): *p*-value > 0.05: There is no significant evidence to reject the null hypothesis that the variances between groups are equal; *p*-value ≤ 0.05: There is significant evidence to reject the null hypothesis of equality of variances (the variances between groups may be different).

Work	Median CV (Cond1) (%)	Median CV (Cond2) (%)	SD del|Log_2_FC (Cond1 vs. Cond2)|	Levene (*p* Value)
1	13.17	12.78	4.28	0.3694
2	26.79	25.50	3.00	0.9991
3	10.16	8.34	2.75	0.0221
4	18.62	31.95	4.72	0.0000
5	36.42	31.99	4.21	0.1310

## Data Availability

The proteomics datasets analyzed in this study are publicly available in the ProteomeXchange repository under the following accession numbers: PXD051640, PXD041209, PXD019139, and PXD034112. All scripts used for statistical analyses, data processing, and meta-analysis are openly available at GitHub (https://github.com/AlfonsoOA/HTM-CRB) and archived in Zenodo (DOI: 10.5281/zenodo.16541357), accessed on 28 July 2025. Versioned computational environments (R package versions, seed values) are provided to ensure reproducibility.

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
