# Peer review of "Reproducibility Crossroads: Impact of Statistical Choices on Proteomics Functional Enrichment"

_ijms, 2025, doi:10.3390/ijms26189232_

Round 1
Reviewer 1 Report (New Reviewer)
Comments and Suggestions for Authors
The manuscript presents a well-structured and timely study addressing a critical aspect of proteomics research: the reproducibility of functional enrichment analyses when applying different statistical methods and biological relevance criteria. The use of five independent real-world datasets, analyzed with both frequentist and Bayesian approaches, is a clear strength that increases the robustness and generalizability of the conclusions. The finding that biological relevance criteria have a greater impact than the choice of statistical test is novel and of high practical importance, and it will certainly be of interest to researchers working in proteomics and omics reproducibility.
Strengths include the originality of analyzing real-world datasets instead of benchmarked ones, the careful application of multiple statistical approaches, and the use of robust concordance metrics (Jaccard, correlations, Euclidean similarity). The manuscript is also clearly within the scope of IJMS.
That said, some aspects could be improved to increase clarity and impact:
English editing: While the language is generally strong, some sentences are overly long and dense. Shortening them would improve readability. Occasional redundancies and calque-like expressions (“in this sense”) should be polished.
Figures: Some plots, particularly the UpSet and boxplots, are visually dense. More explanatory figure legends would greatly help readers interpret them.
Discussion: The discussion would benefit from a slightly broader contextualization. Linking your findings to reproducibility frameworks (e.g., FAIR principles, MIAPE, HUPO guidelines) and providing concrete examples of biological pathways where enrichment interpretation diverged would make the work even stronger.
Comments on the Quality of English Language- Divide overly long sentences into shorter, more direct statements.
- Remove redundant connectors (however, conversely, in this sense).
- Replace calque-like phrases with more standard English (e.g., “in this sense” → “accordingly”).
- Ensure consistent verb tense (past tense for methods/results; present tense for general statements).
- Apply a light professional language edit (minor English editing) to polish flow and readability.
Author Response
Comment 1: English editing: While the language is generally strong, some sentences are overly long and dense. Shortening them would improve readability. Occasional redundancies and calque-like expressions (“in this sense”) should be polished.
Response 1: We thank the reviewer for the positive assessment of our study. We acknowledge that some sentences in the manuscript are long and may reduce readability. We have carefully revised the text to shorten, by splitting, overly complex sentences in Introduction, Methods, and Discussion. Indeed, we eliminated redundancies, avoid expressions such as “in this sense” and replaced them with clearer connectors (e.g., ‘therefore’)
Comment 2: Figures: Some plots, particularly the UpSet and boxplots, are visually dense. More explanatory figure legends would greatly help readers interpret them.
Response 2: Thank you for your comment. We agree with the reviewer that some figures, particularly the UpSet plots and boxplots, may appear visually dense. To address this, we modified figure legends for figures 2 and 4. Indeed, we reviewed the figure caption of figure 5 to confirm that provides both raw and transformed views of the similarity metrics, which facilitates interpretation of global trends and robustness across Works.
Comment 3: Discussion: The discussion would benefit from a slightly broader contextualization. Linking your findings to reproducibility frameworks (e.g., FAIR principles, MIAPE, HUPO guidelines) and providing concrete examples of biological pathways where enrichment interpretation diverged would make the work even stronger.
We appreciate this suggestion. In the revised Discussion, we have included a broader contextualization of our findings. Specifically, we link our results to reproducibility frameworks such as the FAIR principles, MIAPE recommendations, and HUPO guidelines. Additionally, we provide concrete examples of pathways where enrichment interpretations diverged across methods (e.g., metabolic and stress response pathways), illustrating the practical implications of methodological choices. Reviewer can find in current lines 483-505:
“Given the strong impact of methodological choices on biological interpretations, ensuring reproducibility also requires transparent reporting and adherence to established data standards. Adopting the FAIR data principles (Findable, Accessible, Interoperable, Reusable) and reporting standards such as MIAPE and HUPO-PSI would facilitate method comparison and reuse by other groups. Concretely, providing raw inputs to enrichment tools (protein ID lists, parameters, ClueGO xls outputs), code for the statistical pipeline, and versioned environment details (R package versions, seed values) enables direct replication and reduces ambiguity in downstream interpretations.
As concrete examples of divergence, in Work 1 (up-regulated proteins) we observed that some pathways were consistently detected across both filtering strategies. For instance, the KEGG pathway “Complement and coagulation cascades” (hsa04610) and the GO term “acute-phase response” (GO:0006953) were significant under both Bayesian- and FC-based criteria when using DEqMS and MSstats. However, many other terms appeared exclusively in the FC-filtered lists. With MSstats, pathways such as “Protein processing in endoplasmic reticulum” (hsa04141) and “translation” (GO:0006412) were enriched only under FC filtering, whereas the Bayesian approach did not retain them. Similarly, with DEqMS we found disease-related pathways, including “Parkinson’s disease” (hsa05012), to be highlighted only under FC filtering. Notably, no terms were found exclusively after these HTMs with the Bayesian criterion, suggesting that this approach was more conservative. These examples illustrate how the choice of biological relevance criterion can markedly alter the biological interpretation, either by converging on robust terms or by excluding potentially relevant path-ways depending on the filtering strategy.”
Reviewer 2 Report (New Reviewer)
Comments and Suggestions for Authors
The authors made e very detailed and methodologically rigorous study comparing different bioinformatics pipelines applied to each study and then estimating the relative similarities of the results obtained by these pipelines in terms of correlation coefficients (Pearson and Spearman coefficient) and two distance Jaccard (based upon non zero entries) and Euclidean (all data) metrics.
The convergence between different strategies is low and the authors claim for an effort to reduce error sources and adopt a common 'ideal' strategy. While the authors should be complimented for a very rigorous study, in my opinion they completely miss the point that the 'enemy' to face is not noise but the presene of a super-dominant (but not informative) signal that obscures the biologically rlevant information that hides in minor components. This is an old problem in biometrics (see attached file). The super-dominant signal is traditionally called 'size' component (see attached) while the minor information rich components are called 'shape' components. Thus, the only possibility to get rid of the problem is to eliminate size component (NOT TO BE CONFUSED WITH BATCH EFFECT), that is the natural consequence of the fact that there are some proteins that are in any case more abundant than others and that this variability dominates any biological data set, especially if is made of homogeneous material (same cell liners, same tissues). If we wish to get a chance to overcome this problem and obtain a good consistency of the results we do not look for an impossible to attain total elimination of the error sources but we must study the minor components. This can be made as suggested in https://link.springer.com/article/10.1186/1471-2105-7-194 : select components that discriminate the studied conditions (healthy/disease, control/treated) in terms of component scores and then give a 'biological name' to these components by ordering the protein species according to component loadings.
I am conscious this strategy is completely heterogeneous with respect to the authors phylosophy but this is my viewpoint and I had to make it overt to the authors.
https://link.springer.com/article/10.1186/1471-2105-7-194

Author Response
Comment 1: The authors made e very detailed and methodologically rigorous study comparing different bioinformatics pipelines applied to each study and then estimating the relative similarities of the results obtained by these pipelines in terms of correlation coefficients (Pearson and Spearman coefficient) and two distance Jaccard (based upon non zero entries) and Euclidean (all data) metrics.
The convergence between different strategies is low and the authors claim for an effort to reduce error sources and adopt a common 'ideal' strategy. While the authors should be complimented for a very rigorous study, in my opinion they completely miss the point that the 'enemy' to face is not noise but the presene of a super-dominant (but not informative) signal that obscures the biologically rlevant information that hides in minor components. This is an old problem in biometrics (see attached file). The super-dominant signal is traditionally called 'size' component (see attached) while the minor information rich components are called 'shape' components. Thus, the only possibility to get rid of the problem is to eliminate size component (NOT TO BE CONFUSED WITH BATCH EFFECT), that is the natural consequence of the fact that there are some proteins that are in any case more abundant than others and that this variability dominates any biological data set, especially if is made of homogeneous material (same cell liners, same tissues). If we wish to get a chance to overcome this problem and obtain a good consistency of the results we do not look for an impossible to attain total elimination of the error sources but we must study the minor components. This can be made as suggested in https://link.springer.com/article/10.1186/1471-2105-7-194 : select components that discriminate the studied conditions (healthy/disease, control/treated) in terms of component scores and then give a 'biological name' to these components by ordering the protein species according to component loadings.
I am conscious this strategy is completely heterogeneous with respect to the authors phylosophy but this is my viewpoint and I had to make it overt to the authors.
Response: We thank the reviewer for raising this important point. We agree that the dominance of highly abundant proteins can obscure subtle but biologically relevant signals, and we acknowledge that this phenomenon has been extensively discussed in the biometrics literature as the “size component.” Our study focused on evaluating reproducibility across statistical pipelines, keeping the datasets as they are typically analyzed in proteomics practice. While our framework did not explicitly separate “size” and “shape” components, we now explicitly acknowledge in the revised Discussion that this source of variance may contribute to the observed inconsistencies across enrichment results.
We also cite and briefly discuss the suggested reference (Roden et al., 2006), which introduces component-based approaches to disentangle dominant and minor signals. We note that such strategies are complementary to our reproducibility framework and could be an interesting future extension of our work, particularly for datasets where dominant abundance patterns might mask condition-specific biological effects.
Although our study does not directly implement this component-based approach, we believe that recognizing it as an alternative strategy enriches the interpretation of our results and points to a valuable avenue for future research in functional enrichment reproducibility.
Round 2
Reviewer 2 Report (New Reviewer)
Comments and Suggestions for Authors
I wish to compliment the authors for their intellectual honesty of quoting a possible alternative to their approach. So doing, the authors, at least in my opinion, go beyond their methodological frame of reference and suggest possible alternative ways to face the problem of consistency. As often happens when facing complex systems behavior, many problems stem from scaling issues and in many istances the most detailed scale (here the single protein species) is not necessarily the most relevant one (see attached) . This scaling problems are particularly evident in the present work when considering that the major source of inconsistencies comes from the 'biological relevance' side. Now the manuscript is surely worth of publication.

This manuscript is a resubmission of an earlier submission. The following is a list of the peer review reports and author responses from that submission.
Round 1
Reviewer 1 Report
Comments and Suggestions for Authors
The manuscript presents a timely and systematic meta-analysis exploring how different statistical testing strategies and biological relevance criteria influence the outcomes of functional enrichment in label-free quantitative proteomics. While the scope is relevant and the methodology overall sound, several aspects of the design, reporting, and interpretation would benefit from clarification or expansion.
Specific comments:
- The introduction presents a broad rationale for the study; however, it remains somewhat basic and lacks engagement with recent methodological developments. It would be beneficial to reference de novo identification (and more recent search engines like PEAKS) as well as quantitative TMT-based strategies, particularly given their relevance to reproducibility and functional analysis.
- The manuscript focuses exclusively on GO and KEGG. However, other resources may offer broader or complementary insights, especially considering the cross-species nature of the datasets. The authors are encouraged to comment on the rationale behind this limited scope. E.g., the omission of WikiPathways in the functional analysis is noticeable. Do the authors have specific reasons for excluding it? A short justification would be helpful.
- The phrase “more appropriate test” (Table 1) appears to suggest a conclusion derived from an empirical comparison and is more suitable as part of the "results" section, not the "introduction".
- The construction of datasets 4 and 5 is not clearly explained. Specifically, why are these datasets grouped under a single ProteomeXchange ID? Some justification of this approach and its potential effects on the analysis would be appropriate. Furthermore, it would be useful to comment on whether observed between-dataset variability would persist if using five more homogeneous human datasets with comparable proteome coverage.
- The authors mention that all statistical methods were designed to handle missing values. Still, the specific missing data strategies (e.g., imputation, deletion, model-based handling) are not described for each tool. This is critical, given the known sensitivity of proteomics pipelines to assumptions about missing data. Please include a paragraph detailing these strategies per method.
- The manuscript does not clearly state whether Over-Representation Analysis (ORA) or Gene Set Enrichment Analysis (GSEA) was employed (or both). Since these strategies are conceptually distinct and can yield different results, this choice and its justification should be made explicit.
- Line 586, when stating that the method “allowed separating overrepresented and underrepresented categories,” please specify which part of the analytical pipeline this refers to (i.e., the enrichment method, statistical filtering, or ranking procedure).
- The manuscript suggests that Hypothesis Testing Method (HTM) and Criteria for Biological Relevance (CBR) are independent factors. However, in practical workflows, both are commonly used in tandem—e.g., DE analysis frequently includes both FDR control and fold change cutoffs. At the same time, Bayesian models often integrate effect size into significance estimation. Please provide a rationale for treating HTM and CBR as orthogonal, and, if possible, cite examples where only one of these criteria is used.
Minor:
- The overall structure of comparisons across datasets and methods is not immediately apparent from the text. Including a schematic diagram summarizing the study design and analytical comparisons (e.g., what was compared with what, at which step) would improve reader comprehension. Additionally, Figure 1B contains very small font, making it hard to read even when zoomed in. Please consider increasing the font size for accessibility.
- Line 350, "were were" appears duplicated.
Overall, this is a thoughtful and valuable contribution to the field of proteomics. The work raises essential questions about reproducibility and interpretation in functional enrichment analyses, and I appreciate the effort to compare diverse statistical strategies systematically. With some clarifications and additions the manuscript can become even stronger. I look forward to seeing the revised version.
Author Response
Comment 1. The introduction presents a broad rationale for the study; however, it remains somewhat basic and lacks engagement with recent methodological developments. It would be beneficial to reference de novo identification (and more recent search engines like PEAKS) as well as quantitative TMT-based strategies, particularly given their relevance to reproducibility and functional analysis.
Author Response: We appreciate the reviewer's insightful suggestion regarding the introduction. We agree that a broader context on recent methodological advancements is essential for a comprehensive overview of the field.
To address this, we have extensively revised the introduction to incorporate a more detailed discussion of state-of-the-art approaches. The updated text now explicitly references various label-based strategies, including metabolic (SILAC), chemical (iTRAQ), and isobaric (TMT) labeling, and discusses their respective advantages and disadvantages, such as multiplexing capacity and the issue of ratio compression. Furthermore, we have included a new section on data acquisition methods, highlighting the emergence of Data-Independent Acquisition (DIA) as a key development to improve reproducibility and proteome coverage. The revised introduction also now includes a reference to de novo identification and search engines like PEAKS, as suggested, to provide a more complete picture of protein identification workflows.
We believe these revisions provide a more current and robust framework for our study, demonstrating its relevance within the diverse and evolving landscape of quantitative proteomics.
Comment 2. The manuscript focuses exclusively on GO and KEGG. However, other resources may offer broader or complementary insights, especially considering the cross-species nature of the datasets. The authors are encouraged to comment on the rationale behind this limited scope. E.g., the omission of WikiPathways in the functional analysis is noticeable. Do the authors have specific reasons for excluding it? A short justification would be helpful.
Response 2: We appreciate this constructive comment. The decision to focus exclusively on GO and KEGG was deliberate. WikiPathways is just one of many alternatives we could have explored, along with analyses of InterPro terms or subcellular localization, or even the use of different enrichment algorithms like ClusterProfiler. However, our primary goal was to perform a meta-analysis that was as comparable as possible across diverse datasets and species. GO and KEGG are two of the most widely used and consistently maintained databases, providing a robust and standardized framework for functional enrichment analysis. Including additional analyses, while valuable, would have significantly increased the complexity of the study, obscuring our central findings about the impact of statistical decisions. We have added a brief justification to the manuscript, clarifying that our choice was guided by the need for a standardized and widely applicable framework for cross-species comparison, thereby allowing us to more effectively isolate the effects of the statistical methods.
Comment 3. The phrase “more appropriate test” (Table 1) appears to suggest a conclusion derived from an empirical comparison and is more suitable as part of the "results" section, not the "introduction".
Response 3: We thank the reviewer for this valuable feedback. We agree that the phrasing could be misleading. We have revised Table 1, changing the column heading from "More Appropriate Test" to "Other Possible Tests" to avoid suggesting a definitive conclusion in the introduction. We believe retaining this table is important as it provides a broader context for the study, demonstrating that the statistical decisions we investigate are relevant to a wide array of experimental designs beyond the simple A vs. B comparisons explored in this manuscript.
Comment 4. The construction of datasets 4 and 5 is not clearly explained. Specifically, why are these datasets grouped under a single ProteomeXchange ID? Some justification of this approach and its potential effects on the analysis would be appropriate. Furthermore, it would be useful to comment on whether observed between-dataset variability would persist if using five more homogeneous human datasets with comparable proteome coverage.
Response 4: We appreciate the reviewer's request for clarification. The two datasets, Work 4 and Work 5, are grouped under a single ProteomeXchange ID (PXD034112) because they originate from the same original study and were generated from the same set of raw files. However, they represent two distinct pairwise comparisons: Work 4 compares a control condition to a nitrogen fixation condition, while Work 5 compares the same control condition to a phosphorus mobilization condition. We have added a detailed explanation of this to the manuscript to prevent any confusion. In this sense, the paragraph where Work 4 and Work 5 are described have been rewritten as: Works 4 and 5 (ProteomeXchange: PXD034112) were derived from a comprehensive study on biological nitrogen fixation and phosphorus mobilization in Azotobacter chroococcum NCIMB 8003 [36]. For dataset 4, raw files from control and biological ni-trogen fixation conditions were used, while for dataset 5, raw files corresponding to control conditions were compared to phosphorous mobilization conditions. The protein database for these works was UP000068210".
Regarding the question about using more homogeneous human datasets, this is an excellent point for future research. Our primary goal was to assess the impact of statistical choices in a meta-analysis framework, and our intentional use of heterogeneous datasets (animal, plant, and bacterial) was designed to demonstrate that the observed variability due to statistical decisions is a general phenomenon, not restricted to a specific model organism. This approach strengthens the universal applicability of our conclusions. We have added a comment to the discussion acknowledging that a future study using a homogeneous set of datasets would be an interesting avenue to explore the consistency of these effects within a single, complex biological system (at the end of results and discussion section).
Comment 5. The authors mention that all statistical methods were designed to handle missing values. Still, the specific missing data strategies (e.g., imputation, deletion, model-based handling) are not described for each tool. This is critical, given the known sensitivity of proteomics pipelines to assumptions about missing data. Please include a paragraph detailing these strategies per method.
Response 5: We thank the reviewer for this insightful and important comment. In response, we have revised the manuscript to explicitly describe how missing values were handled in each statistical method used in the differential abundance analysis (see revised Section 2.2). Rather than applying global imputation, we opted to respect each method’s native assumptions and internal handling of missingness. Specifically:
- For Student's and Welch’s t-tests, pairwise complete observations were used, meaning proteins with at least two non-missing values per group were retained and no imputation was performed.
- Limma and DEqMS tolerate missing data natively, leveraging empirical Bayes shrinkage for variance estimation provided that sufficient replicates are available, without requiring imputation.
- MSstats applies a model-based approach treating missing values as censored (non-random) data; this avoids imputation and allows valid inference under a missing-not-at-random (MNAR) assumption.
- The Bayesian model implemented via rstanarm handled missing values implicitly through marginalization in the posterior distribution, again avoiding any explicit imputation.
These details have now been included in the manuscript to clarify that each tool was used in a manner consistent with its own statistical assumptions regarding missing data. We believe this clarification strengthens the methodological transparency of our work.
Comment 6. The manuscript does not clearly state whether Over-Representation Analysis (ORA) or Gene Set Enrichment Analysis (GSEA) was employed (or both). Since these strategies are conceptually distinct and can yield different results, this choice and its justification should be made explicit.
Author Response: We apologize for this omission. For the functional enrichment analysis, we used Over-Representation Analysis (ORA). This choice was made because ORA is a well-established method for analyzing discrete lists of differentially abundant proteins, which is precisely the output of our statistical filtering steps. We have clarified this choice in the manuscript to ensure that our methodology is transparent.
Comment 7. Line 586, when stating that the method “allowed separating overrepresented and underrepresented categories,” please specify which part of the analytical pipeline this refers to (i.e., the enrichment method, statistical filtering, or ranking procedure).
Response 7: We thank the reviewer for this valuable comment. The phrase to which you refer relates to a key step in our analytical pipeline. Specifically, after performing hypothesis testing and applying criteria for biological relevance, proteins were separated into lists of upregulated and downregulated proteins, on which the functional enrichment analysis (ORA) was performed. Therefore, the separation of 'overrepresented and underrepresented categories' refers to this initial classification of the protein lists that served as input for the ORA, which was then analyzed globally.
Comment 8. The manuscript suggests that Hypothesis Testing Method (HTM) and Criteria for Biological Relevance (CBR) are independent factors. However, in practical workflows, both are commonly used in tandem—e.g., DE analysis frequently includes both FDR control and fold change cutoffs. At the same time, Bayesian models often integrate effect size into significance estimation. Please provide a rationale for treating HTM and CBR as orthogonal, and, if possible, cite examples where only one of these criteria is used.
Response 8: The reviewer raises an excellent and very important point about the interplay between HTMs and CBRs. We acknowledge that in real-world workflows, these two are often combined. Our decision to treat them as orthogonal factors in this study was a deliberate part of our experimental design, aiming to deconstruct and quantify their individual contributions to the final results. By systematically analyzing the output of different HTMs with the application of different CBRs, we could isolate the specific impact of each step. For example, we first compared the lists of significant proteins obtained from each HTM using only a p-value cutoff, and then we introduced fold-change or Bayesian-based cutoffs (CBR) to see how the overlap and concordance of these lists changed. This systematic deconstruction allowed us to provide a granular view of how each decision influences the outcome.
Minor Comments
MC1. The overall structure of comparisons across datasets and methods is not immediately apparent from the text. Including a schematic diagram summarizing the study design and analytical comparisons (e.g., what was compared with what, at which step) would improve reader comprehension. Additionally, Figure 1B contains very small font, making it hard to read even when zoomed in. Please consider increasing the font size for accessibility.
Response MC1: We agree with both points. We have developed a schematic diagram that visually summarizes our study design and analytical pipeline, which will be included in the revised manuscript to improve clarity. We have also addressed the readability of Figure 1B by increasing the font size to make it more accessible.
MC2. Line 350, "were were" appears duplicated.
Response MC2: Thank you for highlighting this error, this has been fixed.
Reviewer 2 Report
Comments and Suggestions for Authors The processing and analysis of proteomic data is an essential step, as it directly impacts the results presented and the biological analysis of the data obtained. Therefore, it is an important field of study addressed in this manuscript. In this work, the authors compare different statistical analyses on data from five works deposited in ProteomeXchange, highlighting the impact on functional enrichment (GO and KEGG). My main concern is that the first result (total number of significant proteins identified) using six hypothesis testing methods (Bayesian, DEqMS, Limma, MSstats, t-Student, and t-Welch) presents a very discrepant number of proteins in Work 2, 3, and 5. This impacts all subsequent results, such as Biological Relevance in Differential Proteomics Analysis. With paper 3, it's not even possible to perform all the analyses. Overall, differences between the gene ontology analyses are expected. In this reviewer's opinion, the article's proposal is very interesting and relevant to the literature, but the way it was carried out does not allow for supporting the conclusions. Questions: 1. The title needs to be changed to "Impact of Statistical Choices on Proteomics Functional Enrichment." Since there is a marked difference in the number of significant proteins using different hypothesis testing methods (HTMs), an impact on functional enrichment is expected. This statement is only possible if the authors present only work in which the number of proteins is not so distinct, but functional differences appear. 2. The three criteria for selecting Work 1, 2, and 3 yielded approximately 30 dataset deposited on ProteomeXchange. From there, were the data randomly selected? 3. The authors compared studies involving animal (Mus musculus), plant (Arabidopsis thaliana), and bacterial (Escherichia coli) cells, as well as the group model (Azotobacter chroococcum). Since these models are very different, couldn't this have affected the analyses? For example, the authors could have selected only studies involving plant species or even Mus musculus or Homo sapiens (which corresponds to the largest data set of the three proposed criteria). Furthermore, the parameters adopted for the analyses were the same, and this could have affected the analysis of such different modulations. 4. Lines 157-159: "Works 4 and 5 (ProteomeXchange: PXD034112) were derived from a comprehensive study on biological nitrogen fixation and phosphorus mobilization in Azotobacter chroococcum NCIMB 8003 [36]." The article does not make clear the difference between Work 4 and 5, since both have a single reference and a deposit in the protein database. 5. The authors characterized dataset variability using the median coefficient of variation (CV). Did performed Principal Component Analysis (PCA) on the data from each Work? PCA is a data analysis method used to reduce data dimensionality and identify patterns, allowing visualization of the differences between the conditions evaluated in each Work. 6. "Table 1. Recommended Statistical Tests for Quantitative Proteomics Differential Expression Analysis Based on Experimental Design". Table 1 should not be included in the Introduction, as it discusses analyses. Furthermore, in the manuscript, the authors discuss five works that only consider "Simple Comparison (A vs. B)." The sentences about this table could primarily address the question addressed in the manuscript's results. Suggestions 1. Abstracts should not have paragraphs. 2. Lines 47-48: "Unlike isotopic or chemical labeling methods, it infers relative protein abundance directly from peptide ion signal intensity in the mass spectrometer." The authors need to correct this sentence by citing a reference since in label-free quantitative proteomics, intensity-based methods and spectral counting methods are two distinct approaches for determining protein abundance. 3. Lines 68-69: The authors discuss issues prior to the development of equipment for analysis. The points raised constitute limitations even for labeling methods in quantitative proteomics. Furthermore, the authors do not explore the issues of mass spectrometers, jumping directly to the analysis of the results. The equipment used is essential for label-free methods. 4. Line 126: "GO, KEGG, and Reactome." It is necessary to add the full name of GO and KEGG, this being the first time they appear in the text. 5. Line 151: " Work 2 (ProteomeXchange: PXD041209) investigated the Escherichia coli protein" and Line 156: " database was Arabidopsis thaliana (UP000006548).". These scientific names must be italicized.Author Response
Reviewer 2
Comment 1: The title needs to be changed to "Impact of Statistical Choices on Proteomics Functional Enrichment." Since there is a marked difference in the number of significant proteins using different hypothesis testing methods (HTMs), an impact on functional enrichment is expected. This statement is only possible if the authors present only work in which the number of proteins is not so distinct, but functional differences appear.
Response 1: Thank you for pointing this out. However, we respectfully disagree with the reviewer's suggestion to change the title. Our work directly demonstrates that both the choice of hypothesis testing and the method for determining biological relevance influence functional enrichment analyses. We believe the current title sufficiently captures both types of sources of variation.
Comment 2: The three criteria for selecting Work 1, 2, and 3 yielded approximately 30 dataset deposited on ProteomeXchange. From there, were the data randomly selected?
Response 2: Thank you for your comment. Yes, the datasets for Works 1, 2, and 3 were selected randomly from the pool of approximately 30 datasets that met the specified criteria (Orbitrap Fusion mass spectrometer, DDA mode, and peer-reviewed publication). This random selection was performed to ensure that our meta-analysis was not biased towards any specific biological system or experimental design, thereby strengthening the generalizability of our findings regarding the impact of statistical choices.
Comment 3: The authors compared studies involving animal (Mus musculus), plant (Arabidopsis thaliana), and bacterial (Escherichia coli) cells, as well as the group model (Azotobacter chroococcum). Since these models are very different, couldn't this have affected the analyses? For example, the authors could have selected only studies involving plant species or even Mus musculus or Homo sapiens (which corresponds to the largest data set of the three proposed criteria). Furthermore, the parameters adopted for the analyses were the same, and this could have affected the analysis of such different modulations.
Response 3: We appreciate this comment, but the intention of this study was to analyze how statistical decisions affect a series of proteomic datasets, irrespective of their biological origin. We considered that selecting more similar datasets would not alter the core conclusions of the present work. As shown in the results, even in complex datasets like Work 1 (Mus musculus) and Work 3 (Arabidopsis thaliana), significant differences emerge depending on the choice of hypothesis testing and biological relevance criteria. The use of diverse organisms, from prokaryotic to eukaryotic, across different kingdoms (Animalia and Plantae), underscores the robustness and broad applicability of our findings. This diversity reinforces the central message that the methodological choices in the bioinformatics pipeline have a universal impact on the final results, regardless of the biological system being studied.
Comment 4: Lines 157-159: "Works 4 and 5 (ProteomeXchange: PXD034112) were derived from a comprehensive study on biological nitrogen fixation and phosphorus mobilization in Azotobacter chroococcum NCIMB 8003 [36]." The article does not make clear the difference between Work 4 and 5, since both have a single reference and a deposit in the protein database.
Response 4: Thank you for this comment, which highlights a lack of clarity in our original manuscript. For the present study, Work 4 and Work 5 were generated by performing two distinct pairwise comparisons from the same original study (ProteomeXchange: PXD034112). Specifically, for Work 4, we compared the RAW files from the control condition against the biological nitrogen fixation condition. For Work 5, we compared the RAW files from the control condition against the phosphorous mobilization condition. To improve clarity, we have modified the manuscript to state the following: "Works 4 and 5 (ProteomeXchange: PXD034112) were derived from a comprehensive study on biological nitrogen fixation and phosphorus mobilization in Azotobacter chroococcum NCIMB 8003 [36]. For dataset 4, raw files from control and biological nitrogen fixation conditions were used, while for dataset 5, raw files corresponding to control conditions were compared to phosphorous mobilization conditions. The protein database for these works was UP000068210."
Comment 5: The authors characterized dataset variability using the median coefficient of variation (CV). Did performed Principal Component Analysis (PCA) on the data from each Work? PCA is a data analysis method used to reduce data dimensionality and identify patterns, allowing visualization of the differences between the conditions evaluated in each Work.
Response 5: Thank you for your comment. While PCA is a commonly used method for exploratory data analysis in omics studies, we did not include it in this meta-analysis because our primary objective was to investigate the downstream impact of different statistical decisions on the final functional enrichment results. The original publications for the selected datasets already included exploratory analyses, and our work builds upon those by focusing specifically on the consequences of different hypothesis testing and biological relevance criteria. We believe that incorporating PCA at this stage would not have provided additional insight into the central question of our study.
Comment 6: "Table 1. Recommended Statistical Tests for Quantitative Proteomics Differential Expression Analysis Based on Experimental Design". Table 1 should not be included in the Introduction, as it discusses analyses. Furthermore, in the manuscript, the authors discuss five works that only consider "Simple Comparison (A vs. B)." The sentences about this table could primarily address the question addressed in the manuscript's results.
Response 6: We appreciate this feedback. We have modified Table 1, changing the heading "More Appropriate Test" to "Other Possible Tests" to better reflect the purpose of the table. We believe it is important to keep this table in the Introduction for two key reasons. First, it provides a broader context for the reader, illustrating that the challenges of statistical choices are not limited to simple A vs. B comparisons but are relevant to a wide range of experimental designs in proteomics. Second, our study aims to highlight the general "reproducibility crossroads" in proteomics, and showing the diversity of statistical options available for different designs strengthens this overarching message. This helps to position our work not just as a narrow analysis of A vs. B comparisons, but as a foundational study on the broader implications of statistical decision-making in the field.
Suggestion 1: Abstracts should not have paragraphs.
Response Suggestion 1: We agree with this comment. The abstract has been revised to be a single paragraph, eliminating the separation into multiple paragraphs.
Suggestion 2: Lines 47-48: "Unlike isotopic or chemical labeling methods, it infers relative protein abundance directly from peptide ion signal intensity in the mass spectrometer." The authors need to correct this sentence by citing a reference since in label-free quantitative proteomics, intensity-based methods and spectral counting methods are two distinct approaches for determining protein abundance.
Response Suggestion 2: The reviewer is correct. This sentence requires a clarification and a reference. The original phrasing was an oversimplification. Label-free quantitative proteomics indeed includes both intensity-based and spectral counting methods. Our study, however, is exclusively focused on intensity-based quantification, which we have used with the LFQ algorithm within MaxQuant. We have revised the introduction section taking into consideration this suggestion.
Suggestion 3: Lines 68-69: The authors discuss issues prior to the development of equipment for analysis. The points raised constitute limitations even for labeling methods in quantitative proteomics. Furthermore, the authors do not explore the issues of mass spectrometers, jumping directly to the analysis of the results. The equipment used is essential for label-free methods.
Response Suggestion 3: We thank the reviewer for this suggestion. However, we believe that the issues related to the mass spectrometer and the instrumental stage are already addressed in the introduction. In the paragraph discussing sources of variability (current lines 84-89), we explicitly mention that "The instrumental stage influences by ionization efficiency and peptide ion mobility, among others." We further detail the importance of factors such as "Optimization of mass spectrometer acquisition parameters... co-elution, interference, and the quality/stability of the liquid chromatogram..." and state that "Poor chromatographic reproducibility or high background noise severely compromise accurate peptide quantification." We believe that including these details is crucial, as they highlight the multitude of factors that affect data quality and, consequently, reproducibility in proteomics, which is the central focus of our manuscript.
Suggestion 4: Line 126: "GO, KEGG, and Reactome." It is necessary to add the full name of GO and KEGG, this being the first time they appear in the text.
Response Suggestion 4: We agree. The full names have been added upon their first appearance in line 127. The text now reads: "GO (Gene Ontology), KEGG (Kyoto Encyclopedia of Genes and Genomes), and Reactome. This provides..."
Suggestion 5: Line 151: "Work 2 (ProteomeXchange: PXD041209) investigated the Escherichia coli protein" and Line 156: "database was Arabidopsis thaliana (UP000006548).". These scientific names must be italicized.
Response Suggestion 5: We agree with the reviewer's suggestion. The scientific names have been italicized throughout the manuscript for consistency and adherence to standard scientific formatting.
Round 2
Reviewer 1 Report
Comments and Suggestions for Authors
Dear authors,
Thank you for addressing all the points raised during the review process. The revised manuscript has been improved accordingly. Further validation of your results across multiple datasets within a single organism would strengthen the study’s reliability. I encourage you to pursue this in future work.
Author Response
Comment 1. "Dear authors, Thank you for addressing all the points raised during the review process. The revised manuscript has been improved accordingly. Further validation of your results across multiple datasets within a single organism would strengthen the study’s reliability. I encourage you to pursue this in future work."
Response 1: Dear Reviewer 1,
We are truly grateful for your constructive feedback and positive assessment of our revised manuscript. We are pleased to hear that you found the manuscript improved. Your valuable comments throughout the review process have significantly contributed to enhancing the overall quality and clarity of the paper.
We appreciate your suggestion regarding further validation of our results across multiple datasets within a single organism. This is an insightful point, and we certainly intend to explore this in our future work to strengthen the study's reliability even further.
Thank you once again for your time and thoughtful contributions.
Reviewer 2 Report
Comments and Suggestions for Authors
The authors responded to the questions, but this reviewer continues to disagree with the methodology. When I referred to the PCA (Principal Component Analysis) data, which was also performed in the group's previous manuscript (ref. 27), I was referring to the differences in a pairwise comparison between the close (C and PM) and more distant (C and BNF) conditions. Respectfully, my criticism is not of the manuscript's proposal, but of the quantity and characteristics of the data selected for analysis.
Author Response
Comment 1: "The authors responded to the questions, but this reviewer continues to disagree with the methodology. When I referred to the PCA (Principal Component Analysis) data, which was also performed in the group's previous manuscript (ref. 27), I was referring to the differences in a pairwise comparison between the close (C and PM) and more distant (C and BNF) conditions. Respectfully, my criticism is not of the manuscript's proposal, but of the quantity and characteristics of the data selected for analysis."
Response 1: Dear Reviewer 2,
We appreciate you taking the time to review our revised manuscript and for your continued engagement with our methodology. Regarding your query about Principal Component Analysis (PCA) and the pairwise comparisons, we have now performed PCA on the filtered data from each dataset, addressing your specific points about differences between conditions like C and PM, and C and BNF. The results of these analyses, including an assessment of sample clustering and PERMANOVA statistics, are now incorporated into Figure S1 of the revised manuscript.
We also wish to clarify that the datasets reanalyzed in this study are real-world, previously published data, which inherently represent the diverse range of qualities and characteristics encountered in authentic research. Our meta-analytical approach aims to understand the impact of statistical decisions across this real-world variability. Furthermore, our original analytical approach included a Non-metric Multidimensional Scaling (NMDS) and ANOSIM analysis, which provide complementary insights into multivariate sample dissimilarities. The R script for this analysis, which further demonstrates our comprehensive methodological approach, is openly accessible on GitHub at: https://github.com/AlfonsoOA/HTM-CRB/blob/main/Script1_FilteringAndVisualization.R.
We hope these additions and clarifications address your concerns regarding the analysis and characterization of the selected data.
Thank you for your valuable feedback.